# Individuality in the hive - Learning to embed lifetime social behavior of honey bees

## Abstract

Honey bees are a popular model for complex social systems, in which global behavior emerges from the actions and interactions of thousands of individuals. While the average life of a bee is organized as a sequence of tasks roughly determined by age, there is substantial variation at the individual level. For example, young bees can become foragers early in life, depending on the colony's needs. Using a unique dataset containing lifetime trajectories of all individuals over multiple generations in two honey bee colonies, we propose a new temporal matrix factorization model that jointly learns the average developmental path and structured variations of individuals in the social network over their entire lives. Our method yields inherently interpretable embeddings that are biologically plausible and consistent over time, which allows comparing individuals regardless of when or in which colony they lived. Our method provides a quantitative framework for understanding behavioral heterogeneity in complex social systems applicable in fields such as behavioral biology, social sciences, neuroscience, and information science.

## 1 Introduction

Animals living in large groups often coordinate their behaviors, resulting in emergent properties at the group level, from flocking birds to democratic elections. In most animal groups, the role an individual plays in this process is thought to be reflected in the way it interacts with group members. Technological advances have made it possible to track all individuals and their interactions in animal societies, ranging from social insects to primate groups (Mersch et al., 2013; Gernat et al., 2018; Mathis et al., 2018; Graving et al., 2019; Pereira et al., 2019). These datasets have unprecedented scale and complexity, but understanding these data has emerged as a new and challenging problem in itself (Pinter-Wollman et al., 2014; Krause et al., 2015; Brask et al., 2020).

A popular approach to understand high-dimensional data is to learn semantic embeddings (Frome et al., 2013; Asgari & Mofrad, 2015; Camacho-Collados & Pilehvar, 2018; Nelson et al., 2019). Such embeddings can be learned without supervision, are interpretable, and are useful for accomplishing downstream tasks. Individuals in animal societies can be described with semantic embeddings extracted from social interaction networks using matrix factorization methods. For example, in symmetric non-negative matrix factorization (SymNMF), the dot products of any two animals' factor vectors reconstruct the interaction matrix (Wang et al., 2011; Shi et al., 2015), see Figure 1 a and b). If the embeddings allow us to predict relevant behavioral properties, they serve our understanding as *semantic* representations. However, in temporal settings where the interaction matrices change over time, there is no straightforward extension of this algorithm. The interaction matrices at different time points can be factorized individually, but there is no guarantee that the embeddings stay semantically consistent over time, i.e. the prediction of relevant behavioral properties will deteriorate.

In living systems, interaction dynamics are highly variable; individuals differ in when they appear in the data and how long they live. Different non-overlapping groups of individuals, e.g. from different years, may not interact with each other at all. How can we find a common semantic embedding even in these extreme cases? How do we learn embeddings that generalize to different groups and still provide insights into each individual's functional role? If animals take on roles partially determined by a common factor, such as age, how can we learn this dependency?

Several approaches to extend NMF to temporal settings have been proposed in a variety of problem settings. Yu et al. (2016) and Mackevicius et al. (2019) propose a factorization method for time

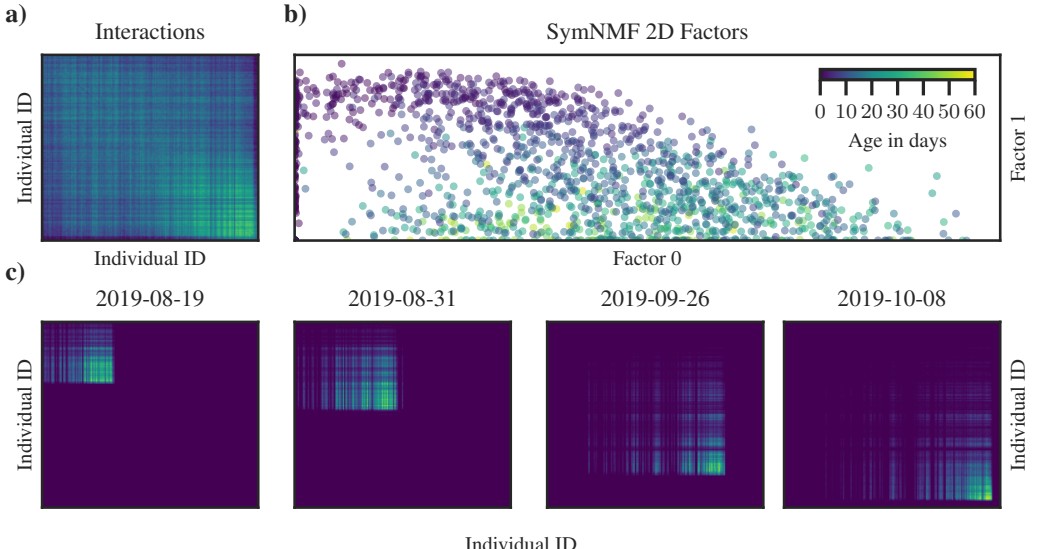

Figure 1: For a daily snapshot of a temporal social network, symmetric NMF is able to extract meaningful factor representations of the individuals. Colors represent the interaction frequencies of all individuals (**a**). The age-based division of labor in a honey bee colony is clearly reflected in the two factors - same-aged individuals are likely to interact with each other (**b**). For long observation windows spanning several weeks, the social network changes drastically as individuals are born, die, and switch tasks (**c**). Here, we investigate how a representation of temporal networks can be extracted, such that the factors representing individuals can be meaningfully compared over time, and even across datasets.

series analysis. Gauvin et al. (2014) focus on the analysis of communities that are determined by their temporal activity patterns. Jiao et al. (2017) consider the case of communities from graphs over time and enforce temporal consistency with an additional loss term. Yu et al. (2017) and Wang et al. (2017) represent networks as a function of time. Temporal matrix factorization can be seen as a tensor decomposition problem, for which methods with many applications have been proposed in the literature, see Kolda & Bader (2009) for a review. In particular, time-shifted tensor decomposition methods have been used in multi-neuronal spike train analysis when recordings of multiple trials from a population of neurons are available (Mørup et al., 2008; Williams, 2020). However, to our knowledge, no method yet considers the case of entities following a common trajectory depending on an observable property (e.g., the age of an individual), which we show to be a powerful inductive bias.

We approach this question using honey bees, a popular model system for studying individual and collective behavior (Elekonich & Roberts, 2005). Bees allocate tasks across thousands of individuals without central control, using an age-based system: young bees care for brood, middle-aged bees perform within-nest labor, and old bees forage outside (Seeley, 1982; Johnson, 2010). Colonies are also organized spatially: brood is reared in the center, honey and pollen are stored at the periphery, and foragers offload nectar near the exit. Therefore, an individual's role is partially reflected in its location, which allows us to evaluate whether the embeddings our method learns are semantically meaningful.

We propose jointly learning two meaningful representations of honey bee social behavior: 1) an *individuality embedding* that characterizes the lifetime behavior of each individual and 2) a daily representation of the individual's functional position in the social network that can be derived from the *individuality embeddings*. We show that these representations can be learned in an unsupervised fashion, using only interaction matrices of the individuals over time. We analyze a dataset obtained by tracking thousands of individually marked honey bees in two colonies, at high temporal and spatial resolution over a total of 155 days, covering entire lifespans and multiple generations. We evaluate how well the embeddings capture the semantic differences of individual honey bee development by evaluating their predictiveness for different tasks and behaviorally relevant metrics.

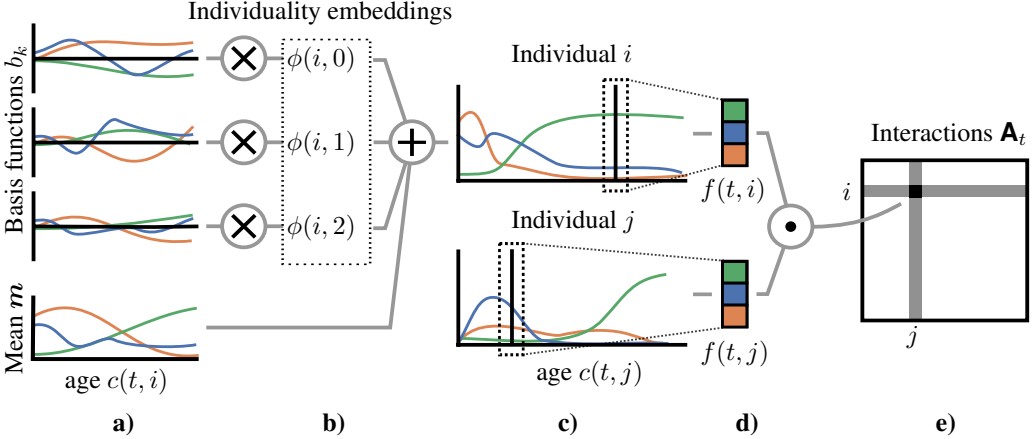

Figure 2: Overview of the method: We learn a parametric function describing the *mean life trajectory* $\boldsymbol{m}(c(t,i))$ and a set of basis functions of individual variation $\boldsymbol{b}(c(t,i))$, where $c(t,i)$ is the age of individual $i$ at time $t$ (**a**). For each individual, an embedding is learned consisting of one scalar per basis function that scales the contribution of the respective basis function - this vector of weights makes up the *individuality embedding* of an individual (**b**). The mean trajectory $\boldsymbol{m}(c(t,i))$ plus a weighted sum of the basis functions $\boldsymbol{b}(c(t,i))$ constitute the *lifetime trajectory* of each individual (**c**). At each time point, factors can be extracted from the individual lifetime trajectories (**d**) to reconstruct the interaction affinity between individuals (**e**). Note that the lifetime trajectories are functions of the individuals' ages, while interactions can occur at any time $t$.

Our method can be used to study dynamic groups in which individual units change their latent properties over time, and group composition is not fixed due to birth, death, or migration of individuals. We provide a new perspective on how to extract fundamental factors that underlie behavioral and developmental plasticity in animal groups.

## 2 METHODS

### 2.1 TEMPORAL NMF ALGORITHM

SymNMF factorizes a matrix $\boldsymbol{A} \in \mathbb{R}_+^{N \times N}$ such that it can be approximated by the product $\boldsymbol{F}\boldsymbol{F}^T$, where $\boldsymbol{F} \in \mathbb{R}_+^{N \times M}$ and $M \ll N$:

$$\hat{\boldsymbol{F}} = \underset{\boldsymbol{F} \geq 0}{\operatorname{argmin}} \left\| \boldsymbol{A} - \boldsymbol{F}\boldsymbol{F}^T \right\|^2 \qquad \boldsymbol{A}_{i,j} \approx \boldsymbol{f}(i) \cdot \boldsymbol{f}(j)^T \qquad \boldsymbol{f}(i) = \boldsymbol{F}_{i,:} \quad \boldsymbol{f}(i) \in \mathbb{R}_+^M \quad (1)$$

When applied to social networks, $\boldsymbol{f}(i)$ can represent the role of an entity within the social network $\boldsymbol{A}$ (Wang et al., 2011; Shi et al., 2015) - however, in temporal settings, factorizing the matrices for different times separately will result in semantically inconsistent factors $\boldsymbol{F}$.

Here we present a novel temporal NMF algorithm (*TNMF*) which extends SymNMF to temporal settings in which $\mathbf{A} \in \mathbb{R}_+^{T \times N \times N}$ changes over time $t$. We assume that the entities $i \in \{0, 1, \ldots, N\}$ follow to some extent a common trajectory depending on an observable property (for example the age of an individual). We represent an entity at a specific point in time $t$ using a factor vector $\boldsymbol{f}^+(t,i)$ such that

$$\hat{\mathbf{A}}_{t,i,j} = \boldsymbol{f}^+(t,i) \cdot \boldsymbol{f}^+(t,j)^T \qquad \hat{\mathbf{A}} \in \mathbb{R}_+^{T \times N \times N} \quad \boldsymbol{f}^+(t,i) \in \mathbb{R}_+^M \quad (2)$$

In contrast to SymNMF, we do not directly factorize $\mathbf{A}_t$ to find the optimal factors that reconstruct the matrices. Instead, we decompose the problem into learning an average trajectory of factors $\boldsymbol{m}(c(t, i))$ and structured variations from this trajectory $\boldsymbol{o}(t, i)$ that depend on the observable property $c(t, i)$:

$$\boldsymbol{f}(t, i) = \boldsymbol{m}(c(t, i)) + \boldsymbol{o}(t, i) \qquad \boldsymbol{f}^+(t, i) = \max(0, \boldsymbol{f}(t, i)) \tag{3}$$

$$c : \mathbb{N}^{T \times N} \to \mathbb{N} \quad \boldsymbol{m} : \mathbb{N} \to \mathbb{R}_+^M \quad \boldsymbol{o} : \mathbb{N}^{T \times N} \to \mathbb{R}^M$$

This decomposition is an inductive bias that allows the model to learn semantically consistent factors for entities, even if they do not share any data points (e.g., there is no overlap in their interaction partners), as long as the relationship between functional role and $c(t, i)$ is stable. Note that in the simplest case $c(t, i) = t$, *TNMF* can be seen as a tensor decomposition model, i.e. the trajectory of all entities is aligned with the temporal dimension $t$ of $\mathbf{A}$. In our case, $c(t, i)$ maps to the age of individual $i$ at time $t$.

While many parameterizations for the function $\boldsymbol{o}(t, i)$ are possible, we only consider one particular case in this work: We learn a set of *individuality basis functions* $\boldsymbol{b}(c(t, i))$ (shared among all entities) that define a coordinate system of possible individual variations and the *individuality embeddings* $\phi$, which capture to what extent each basis function applies to an entity:

$$\boldsymbol{o}(t, i) = \sum_{k=0}^{K} \phi_{i,k} \cdot b_k(c(t, i)) \qquad \phi : \mathbb{R}^{N \times K} \quad b_k : \mathbb{N}^T \to \mathbb{R} \tag{4}$$

where $K$ is the number of learned basis functions. This parameterization allows us to disentangle the forms of individual variability (*individuality basis functions*) and the distribution of this variability (*individuality embeddings*) in the data.

We implement the functions $\boldsymbol{m}(c(t, i))$ and $\boldsymbol{b}(c(t, i))$ with small fully connected neural networks with non-linearities and several hidden layers. The parameters $\theta$ of these functions and the entities' embeddings $\phi$ are learned jointly using minibatch stochastic gradient descent:

$$\hat{\theta}, \hat{\phi} = \underset{\theta, \phi}{\operatorname{argmin}} \left\| \boldsymbol{A} - \hat{\boldsymbol{A}} \right\|^2 \tag{5}$$

Note that non-negativity is not strictly necessary, but we only consider the non-negative case in this work for consistency with prior work (Wang et al., 2011; Shi et al., 2015). Furthermore, instead of one common property with discrete time steps, the factors could depend on multiple continuous properties, i.e. $c : \mathbb{R}^{T \times N} \to \mathbb{R}^P$, e.g. the day and time in a intraday analysis of social networks.

## 2.2 REGULARIZATION

Due to the inductive bias, the model performs well without additional explicit regularization in the datasets considered in this work (see Table 2). However, we find that the model's interpretability can be improved using additional regularization terms without significantly affecting its performance. We encourage sparsity in both the number of used factors and individuality basis functions by adding $L_1$ penalties of the mean absolute magnitude of the factors $\boldsymbol{f}(t, i)$ and basis functions $\boldsymbol{b}(c(t, i))$ to the objective. We encourage individuals' lifetimes to be represented with a sparse embedding using an $L_1$ penalty of the learned *individuality embeddings* $\phi$.

We also introduce an optional adversarial loss term to encourage the model to learn embeddings that are semantically consistent over time, i.e. to only represent two entities that were present in the dataset at different times with different embeddings if this is strictly necessary to factorize the matrices $\mathbf{A}$. We jointly train a discriminative network $d(\phi_i)$ that tries to classify the time of the first occurrence of all entities based on their *individuality embeddings* $\phi$. The negative cross-entropy loss of this model is added as a regularization term to equation 5 in a training regime similar to generative adversarial networks (Goodfellow et al., 2014). See appendix A.2.1 for more details and A.4 for an ablation study of the effect the individual regularization terms have on the results of the model.

## 2.3 DATA

### 2.3.1 HONEY BEE DATA

Two colonies of honey bees were continuously recorded over a total of 155 days. Each individual was manually tagged at emergence, so the date of birth is known for each bee. Locations and identities of all honey bees (N=9286) were extracted from the raw images and used to con-

Table 1: Honey bee datasets

| Dataset | Days | Individuals | Interaction pairs |
|---------|------|-------------|-------------------|
| BN16 | 56 | 2443 | 43 174 748 |
| BN19 | 99 | 6843 | 167 366 381 |

struct daily aggregated temporal interaction networks based on the counts of spatial proximity events. The dataset also contains labels that can be used in proxy tasks (see section 2.5) to quantify if the learned embeddings and factors are semantically meaningful and temporally consistent. See table 1, and appendix A.1 for details. We publish the data together with this paper[1].

### 2.3.2 SYNTHETIC DATA

Although some ground truth labels exist for the honey bee datasets (section 2.5), factorizing these data is still fundamentally an unsupervised learning problem. We generated synthetic datasets to evaluate whether the model can identify groups of individuals with common latent factors that determine their interaction frequencies. We model a common lifetime trajectory and groups of individual variation of factors using smoothed Gaussian random walks. We randomly assign individuals to a group with random times of emergence and disappearance from the data and compute interaction matrices by calculating the dot product of the factors of 1024 individuals for 100 simulated days. We then measure how well the *individuality embeddings* $\phi$ of a fitted model correspond to the truth groups using the adjusted mutual information score (Vinh et al., 2009), and the mean squared error between the ground truth factors and the best permutation of the factors $\boldsymbol{f}^+$. We evaluate the model on 128 different random synthetic datasets with increasing Gaussian noise levels in the interaction tensor. See appendix A.1.2 for more details on the generation process.

In both datasets, we define $c(t, i)$ as the age in days of an individual $i$ at time $t$.

### 2.4 BASELINE MODELS

**Symmetric NMF:** We compute the factors that optimally reconstruct the original interaction matrices using the standard symmetric NMF algorithm (Shi et al., 2015; Kuang et al., 2015), for each day separately, using the same number of factors as in the TNMF model.

**Aligned symmetric NMF:** We consider a simple extension of the standard SymNMF algorithm that aligns the factors to be more consistent over time. For each pair of subsequent days, we consider all combinatorial reorderings of the factors computed for the second day. For each reordering, we compute the mean $L_2$ distance of all individuals that were alive on both days. We then select the reordering that minimizes those pairwise $L_2$ distances and greedily continue with the next pair of days until all factors are aligned. Furthermore, we align the factors across colonies (where individuals cannot overlap) as follows: we run this algorithm for both datasets separately and align the resulting factors by first computing the mean embedding for all individuals grouped by their ages. As before, we now select from all combinatorial possibilities the reordering that minimizes the $L_2$ distance between the embeddings obtained from both datasets. See section A.5.1 for pseudo code.

**Tensor decomposition:** We also compare against a constrained non-negative tensor decomposition model with symmetric factors $\boldsymbol{F} \in \mathbb{R}_+^{N \times M}$ and temporal dynamics constrained to the diagonals, i.e. $\mathbf{D} \in \mathbb{R}_+^{T \times M \times M}$ and $\mathbf{D}_t = \text{diag}(\boldsymbol{d}_t)$, $\boldsymbol{d}_t \in \mathbb{R}_+^M$.

$$\hat{\mathbf{A}}_t = \boldsymbol{F}\mathbf{D}_t\boldsymbol{F}^T \tag{6}$$

$$\hat{\boldsymbol{F}}, \hat{\mathbf{D}} = \underset{\boldsymbol{F},\mathbf{D}}{\text{argmin}}\, T^{-1} \sum_{t=0}^{T} \left\|\boldsymbol{A}_t - \hat{\boldsymbol{A}}_t\right\|^2 \tag{7}$$

**Temporal NMF models:** We evaluate variants of the temporal symmetric matrix factorization algorithms proposed by Jiao et al. (2017) and Yu et al. (2017).

---

[1]See Anonymous (2020) for dataset

For the tensor decomposition and temporal NMF baselines, we follow the procedure given above for the *Aligned symmetric NMF* to find the optimal reordering to align the factors obtained by applying models to the two datasets separately.

### 2.5 EVALUATION METRICS

*Reconstruction:* We measure how well the original interaction matrices **A** can be reconstructed from the factors. We do not require the model to reconstruct the interaction matrices as well as possible because we only use the reconstruction as a proxy objective to learn a meaningful representation. Still, a high reconstruction loss could indicate problems with the model, such as excessive regularization.

*Consistency:* We measure to what extent the *individuality embeddings* $\phi$ change over time. For each model, we train a multinomial logistic regression model to predict the source cohort (date of birth) and calculate the area under the ROC curve ($AUC_{cohort}$) using a stratified 100-fold cross-validation with scikit-learn (Pedregosa et al., 2011). The baseline models do not learn an individuality embedding; therefore we compute how well the model can predict the cohort using the mean factor representation of the individuals over their lives. We define consistency as $1 - AUC_{cohort}$ of this linear model. Note that a very low temporal consistency would indicate that the development of individual bees changes strongly between cohorts and colonies, which we know not to be true.

*Mortality and Rhythmicity:* We evaluate how well a linear regression model can predict the mortality (number of days until death) and circadian rhythmicity of the movement ($R^2$ score of a sine with a period of 24 h fitted to the velocity over a three-day window). These metrics are strongly correlated with an individual's behavior (e.g. foragers exhibit strong circadian rhythms because they can only forage during the daytime; foragers also have a high mortality). We follow the procedure given in Wild et al. (2020) and report the 100-fold cross-validated $R^2$ scores for these regression tasks.

For this data, we expect the factors $\boldsymbol{f}^+$ and *individuality embeddings* $\phi$ to be semantically meaningful and temporally consistent if they reflect an individual's behavioral metrics (mortality and rhythmicity) and if they do not change strongly over time (measured in the consistency metric).

## 3 RESULTS

We implemented the model using PyTorch (Paszke et al., 2019) and trained it in minibatches of 256 individuals for 200 000 iterations with the Adam optimizer (Kingma & Ba, 2015). See appendix A.2.3 for the architecture of the learned functions, a precise description of the regularization losses, and further hyperparameters. The code is publicly available [2].

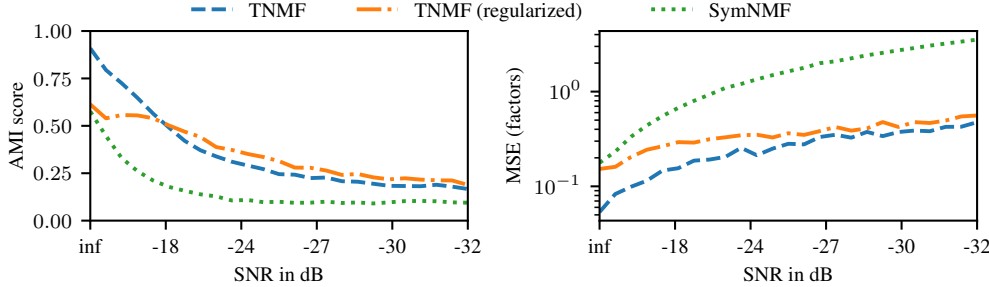

Figure 3: AMI score and mean squared error between true factors and the best permutation of learned factors for increasing noise levels. The median values over 128 trial runs are shown.

### 3.1 SYNTHETIC DATA

We factorize the interaction matrices of the 128 synthetic datasets with varying levels of Gaussian noise. We confirmed that our model converges in all datasets and evaluate whether we can distinguish

---

[2]https://anonymous.4open.science/r/b2b7e2fc-aa04-4cf8-85c7-646d8dc46400

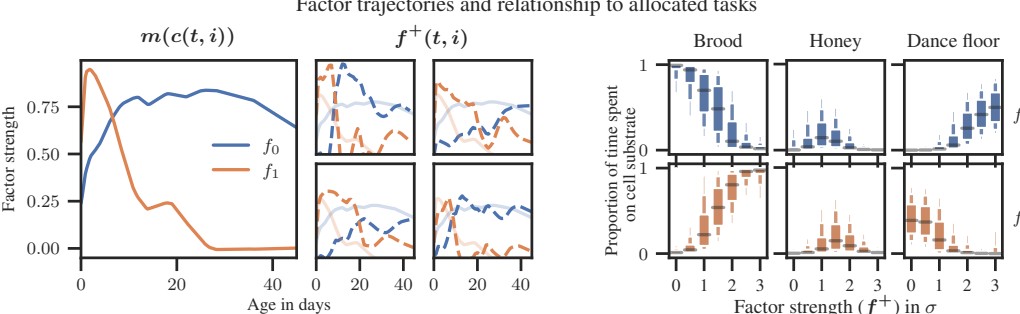

Figure 4: **Left:** Mean lifetime trajectories according to $m(c(t, i))$. The model learns a sparse representation of the functional position of the individuals in the social network. $f_0$ (blue) mostly corresponds to middle-aged and older bees, and $f_1$ (orange) predominantly describes young bees. Only factors with a mean magnitude of at least $0.01$ are shown. Even though the model uses only these two factors, it is still expressive enough to capture individual variability, as can be seen in randomly sampled individuals' lifetime trajectories. **Right:** The individual factors $f^+$ and the proportion of time the individuals spent on different nest substrates. The strong correlation indicates that the learned factors are a good representation of the individuals' roles in the colonies. Note that the factors have been divided by their standard deviation here for ease of comparability.

the individuals' ground truth group assignments. To that end, we extract the *individuality embeddings* $\phi$ from the models and measure how well they correspond to ground truth data using the adjusted mutual information (AMI) score. Furthermore, we measure the mean squared error between the best permutation of learned factors $f^+$ and the ground truth factors. We find that for low levels of noise, our model can identify the truth group assignments with high accuracy, and are still significantly better than random assignments even at very high levels of noise (see figure 3). Note that for this experiment, we evaluated a model with the same hyperparameters as used in all plots in the results section (see Table 2) and a variant without explicit regularization except the $L_1$ penalty of the learned *individuality embeddings* $\phi$ ($\lambda_{\text{embeddings}}$, because this regularization is required to meaningfully extract clusters), which was set to $0.1$. See appendix A.1.2 for more details on the synthetic datasets.

## 3.2 HONEY BEES

**Mean lifetime model:** The model learns a sparse representation of the developmental trajectory of a honey bee in the space of social interactions. Only two factors are effectively used (they exceed the threshold value of $0.01$). These factors show a clear trend over the life of a bee, indicating that the model captures the temporal aspects of the honey bee division of labor (See Figure 4).

**Interpretability of factors:** To understand the relationship between the factors and division of labor, we calculate how the factors map to the fraction of time an individual spent on the brood area, honey storage, or dance floor (where foragers aggregate). Time spent on these different substrates is a strong indicator of an individual's task. The factor $f_1$, which peaks at young age (Figure 4), correlates with the proportion of time spent in the brood area, while a high $f_0$ indicates increased time spent on the dance floor. Therefore, the model learned to map biologically relevant processes.

**Individuality basis functions and individuality embeddings:** Due to the regularization of the embeddings, the model learns a sparse set of *individuality basis functions*. As encouraged by the model, most individuals can predominantly be described by a single basis function. That means that while each honey bee can collect a unique set of experiences, most can be described with a few common *individuality embeddings* which are consistent across cohorts and colonies. In the context of honey bee division of labor, the basis functions are interpretable because the factors correspond to different task groups. For example, $b_{12}(c(t, i))$ (accounting for $\approx 10.7\%$ of the individuals) describes workers that occupy nursing tasks much longer than most bees. As the *individuality embeddings* $\phi$ only scale the magnitude of the basis functions, they can be interpreted in the same way. *Individual lifetime trajectories* in the factor space can be computed based on the *mean lifetime trajectories* ($m$), *individuality basis functions* ($b(c(t, i))$) and *individuality embeddings* ($\phi$). See figure 5 for examples

of *individual lifetime trajectories* from workers that most strongly corresponded to the common *individuality basis functions*.

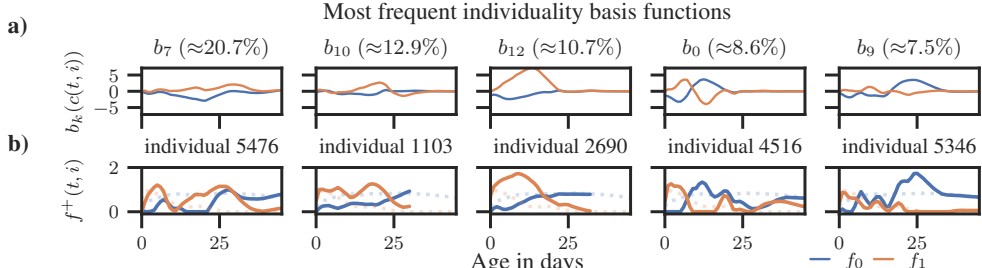

Figure 5: **a)** Magnitude of factor offsets for the five most common individuality basis functions over age $b_k(c(t,i))$. The percentage of individuals that most strongly correspond to the individual basis functions is shown in the column titles. More than 60% of the individuals strongly correspond to one of the five basis functions shown here. **b)** Because the basis functions describe *individuality offsets* from the mean lifetime trajectory, it may be easier to interpret them by visualizing individual examples. For each of the basis functions (top row), we show a lifetime trajectory of an individual that corresponds to that basis function (bottom row). Note that individuals can die or disappear at any time (solid lines). The mean lifetime trajectories are shown as dotted lines in the background.

**Evaluation:** We verify that the learned representations of the individuals are meaningful (i.e., they relate to other properties of the individuals, not just their interaction matrices) and semantically consistent over time and across datasets using the metrics described in the section *Evaluation metrics*. We compare variants of our model with different adversarial loss scaling factors and factor $L_1$ regularizations, the baseline models, and the individuals' ages. We expect a good model to be temporally consistent and semantically meaningful. All variants of our model outperform the baselines in terms of the semantic metrics *Mortality* and *Rhythmicity*, except for the Yu et al. (2017) model, which performs comparably well in the *Mortality* metric. The adversarial loss term further increases the *Consistency* metric without negatively affecting the other metrics. A very strong adversarial regularization (see row with $\lambda_{adv} = 1$ in Table 2) prevents the model from learning a good representation of the data. See Table 2 for an overview of the results. We also evaluate the tradeoff between the different metrics using a grid search over the hyperparameters (see appendix A.3).

**Scalability:** The functions $\boldsymbol{m}(c(t,i))$ and $\boldsymbol{b}(c(t,i))$ are learned neural networks with non-linearities. The objective is non-convex and we learn the model parameters using stochastic gradient descent. Optimization is therefore slower than the standard NMF algorithms that can be fitted using algorithms such as Alternating Least Squares (Kim et al., 2014). We found that the model converges faster if the reconstruction loss of the age based model $\boldsymbol{m}(c(t,i))$ is additionally minimized with the main objective in equation 5. Due to the minibatch training regime, our method should scale well in larger datasets. Small neural networks were sufficient to learn the functions $\boldsymbol{m}(c(t,i))$ and $\boldsymbol{b}(c(t,i))$ in our experiments. Most of the runtime during training is spent on the matrix multiplication $f^+(t,i) \cdot f^+(t,j)^T$ and the corresponding backwards pass.

## 4 CONCLUSION

Temporal NMF factorizes temporal matrices with overlapping and even disjointed communities by learning a common embedding of the lifetime development of the individual entities. In the context of honey bees, this embedding is biologically meaningful, consistent over time, and shows that interaction patterns follow a common lifetime trajectory. Differences from the mean are described in a coordinate system of individual variability. Honey bee colonies are known to exhibit suites of correlated traits (Wray et al., 2011), but this method provides a perspective at the individual level and opens the possibility for understanding how colony-level traits respond to biotic and abiotic pressures. The basis functions are interpretable with respect to the division of labor within colonies and offer a valuable tool to understand and quantify the influence of experimental manipulations (e.g. pesticides or increased predatory pressure) on an individual, even between experimental trials on completely different sets of individuals. While we applied our method to honey bees as an exemplary system

Table 2: Evaluation metrics

|  | Model | | | | |
| Method | Variant | $\left\lVert \boldsymbol{A} - \hat{\boldsymbol{A}} \right\rVert^2$ ↓ | Consistency ↑ | Mortality ↑ | Rhythmicity ↑ |
| --- | --- | --- | --- | --- | --- |
| Age | - | - | - | 0.02 | 0.20 |
| SymNMF | Vanilla | 0.9 | 0.18 | 0.01 | 0.02 |
| SymNMF | Aligned | 0.9 | 0.12 | 0.09 | 0.35 |
| Tensor decomp. | - | 1.36 | 0.03 | 0.06 | 0.09 |
| Jiao et al. (2017) | $\gamma = 0.1$ | 0.9 | 0.19 | 0.02 | 0.05 |
| Jiao et al. (2017) | $\gamma = 1$ | 1.15 | 0.15 | 0.01 | 0.04 |
| Yu et al. (2017) | $\beta = 0.01, d = 5$ | 1.59 | 0.03 | 0.17 | 0.06 |
| **TNMF** | No regularization | 1.21 | 0.17 | 0.30 | 0.48 |
| **TNMF** | $\lambda_{\text{adv}} = 0, \lambda_{\text{f}} = 0.01$ | 1.26 | 0.18 | 0.10 | 0.40 |
| **TNMF** | $\lambda_{\text{adv}} = 0.1, \lambda_{\text{f}} = 0.01$ | 1.28 | 0.35 | 0.20 | 0.42 |
| **TNMF** | $\lambda_{\text{adv}} = 1, \lambda_{\text{f}} = 0.01$ | 1.88 | 0.5 | 0.03 | 0.25 |
| **TNMF** | $\lambda_{\text{adv}} = 0, \lambda_{\text{f}} = 0.1$ | 1.31 | 0.19 | 0.09 | 0.38 |
| **TNMF** | $\lambda_{\text{adv}} = 0.1, \lambda_{\text{f}} = 0.1$ | 1.33 | 0.37 | 0.10 | 0.42 |

Table 3: The evaluation metrics for TNMF and the baseline models described in section 2.5. See appendix A.2.3 and A.5 for descriptions of the hyperparameters used. Note that the SymNMF model reconstruction loss can be seen as a lower bound for the matrix factorization models considered here, and imposing a temporal structure or regularization causes all models to explain less variance in the data. However, for all models except TNMF this does not result in a significant increase of the other metrics. The underlined model is used in all plots in the results section.

Individuality embeddings $\phi$

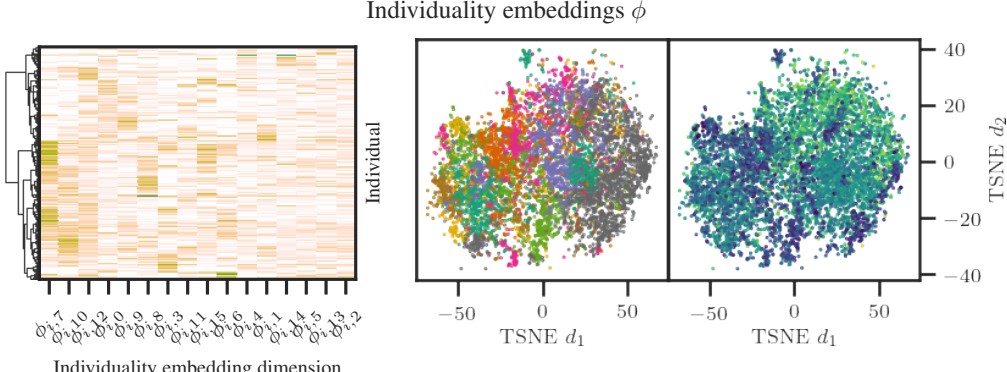

Figure 6: **Left:** Hierarchical clustering of individuality embeddings: Most individuals strongly correspond to a single individuality basis function, making it easy to cluster their lifetime social behavior (i.e. each individual has a high value in a single dimension for their individuality embedding). Because each cluster is strongly associated with a specific individuality basis function, and because each basis function is interpretable (Figure 4), these blueprints of lifetime development can also be intuitively understood and compared. **Right:** TSNE plots of the individuality embeddings colored by cluster (left) and the maximum circadian rhythmicity of an individual during her lifetime (right), indicating that the embeddings are semantically meaningful.

with many individuals that exhibit an entangled, non-overlapping social structure, our method can be applied to any setting in which some interaction structure follows a general pattern over an observable (such as time) to detect structured deviations at the individual level.

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
