# OpenReview forum: "Individuality in the hive - Learning to embed lifetime social behaviour of honey bees"
_ICLR.cc/2021/Conference — Reject_

### Official Review · AnonReviewer4 · 2020-10-28
**Very exciting science and an ambitious solution but lacks important motivation, details, and clarity.**

**Rating:** 6
**Confidence:** 4

**Review:**

The authors present a matrix factorization model to jointly characterize the lifetime interactions of thousands of bees over generations. The problem is fascinating as both a technical and scientific question and the modeling framework appears novel. Although not directly addressed, the authors appear to be trying to solve a *tensor* factorization problem, not just the special case of a matrix (which is of course a 2-d tensor). It would have been interesting to see results comparing their method with a non-negative variant of, say, PARAFAC/CANDECOMP or generalizations thereof. It would have at least have been appropriate to explain why or why not existing tensor methods are not appropriate.

While I'm excited about this work and would love to see the potential applications in other domains, I'm troubled by the lack of clarity and sparsity of essential details in the main text. I am also concerned about how the authors have prioritized the regularization. I detail my concerns below:

Some erroneous (or perhaps just awkward notational issues) make reading difficult and omitted material makes it hard to evaluate the results.
* The authors do not explain in either the main text or in the supplement what $\omega$ is (first appears in equation (3)) and what about it is being learned (as indicated in equation (5)).
* Equation (4) uses $k$ as both an index for the terms in the sum and as a subscript for the number of learned basis functions (although it is not clear why that number has its own index). What did the authors
* How did the authors select the hyperparameters not listed in Table 2?

It seems that the "interpretability" metrics reported in Table 2 are precisely the same criteria as are being regularized for. It therefor should not be surprising that TNMF outperforms SymNMF on these metrics. What is somewhat concerning is how big a difference there is in the reconstruction loss and how much TNMF seems to fail in this regard. This must mean that the features that the authors wish to represent explain very little of the variance in the interaction networks. Could the authors 1) comment on this a bit and discuss why it would be the case and, 2) give us a sense of what the scale of the errors are relative to the total variance of $\mathbf{A}$, 3) comment on whether or not they could be missing high-variance features that they did not anticipate.

Lastly, there is a fundamental problem I have with the authors' regularization approach to interpretability. It seems as though they have enforced a handful of individual features for their model in order to achieve interpretability. However, these features are ostensibly things they already know about the individuals in the population. The point of an unsupervised method is to allow yourself the opportunity to discover things that you didn't already know from observing the features of individuals. Moreover, "interpretability" is somewhat subjective. Therefore, I'm concerned that the authors are using these regularization terms a little too haphazardly and have failed to make a strong argument for their use vis-a-vis their effect on reconstruction error.

I pains me to give this a mediocre score because I do find the work fascinating.  The paper requires more principled motivation for the choices the authors made as well as cleaning up the notation.

---

> ### Author Response · Authors · 2020-11-25
> **Response to Reviewer 4 (1/2)**
>
> > The authors present a matrix factorization model to jointly characterize the lifetime interactions of thousands of bees over generations. The problem is fascinating as both a technical and scientific question and the modeling framework appears novel.
>
> Thank you for the review and the encouraging words! We have revised the manuscript and hopefully addressed your points of criticism:
>
> > Although not directly addressed, the authors appear to be trying to solve a tensor factorization problem, not just the special case of a matrix (which is of course a 2-d tensor). It would have been interesting to see results comparing their method with a non-negative variant of, say, PARAFAC/CANDECOMP or generalizations thereof. It would have at least have been appropriate to explain why or why not existing tensor methods are not appropriate.
>
> We now mention tensor decomposition methods in the introduction and have also included the constrained nonnegative tensor decomposition model proposed by R3 in our evaluation. We have also improved the methods section and now show that the model reduces to a tensor decomposition problem only in the special case of $c(t, i) = t$.
>
> > Some erroneous (or perhaps just awkward notational issues) make reading difficult and omitted material makes it hard to evaluate the results. The authors do not explain in either the main text or in the supplement what is (first appears in equation (3)) and what about it is being learned (as indicated in equation (5)). Equation (4) uses as both an index for the terms in the sum and as a subscript for the number of learned basis functions (although it is not clear why that number has its own index). What did the authors
>
> We have improved the notation throughout the manuscript, and in particular in the methods section. $\omega$ in particular was a somewhat awkward notation used in the initial submission and is not part of the revision anymore. We also fixed the notational issue in Equation (4).
>
> > I am also concerned about how the authors have prioritized the regularization. [..] It seems that the "interpretability" metrics reported in Table 2 are precisely the same criteria as are being regularized for. It therefor should not be surprising that TNMF outperforms SymNMF on these metrics. What is somewhat concerning is how big a difference there is in the reconstruction loss and how much TNMF seems to fail in this regard. This must mean that the features that the authors wish to represent explain very little of the variance in the interaction networks. Could the authors 1) comment on this a bit and discuss why it would be the case and, 2) give us a sense of what the scale of the errors are relative to the total variance of , 3) comment on whether or not they could be missing high-variance features that they did not anticipate.
>
> In the revision, we highlight that the regularizations are optional and that the model works well without them because the implicit bias (factors depend on the individuals' ages and structured deviations from the mean developmental trajectory) is already a powerful regularizer.
>
> We also now better distinguish between "interpretability" and "semantic meaning" of the learned factors and embeddings. In particular, the regularizations are mostly intended to improve the model's interpretability (e.g., by using few factors instead of many strongly correlated factors), but the semantic meaning of the learned representations is high even in the unregularized model (see Table 2). Please note that the metrics in Table 2 are not supposed to measure interpretability, and, as far as we can tell, there is no relationship between the regularizations and the "Mortality" and "Rhythmicity" metrics. The adversarial regularization is supposed to improve temporal consistency, but we think it is still valuable to show that the regularization works and compare it to the baseline models.
>
> Regarding the reconstruction loss, we repeat our reply to a similar question from R3 here: The SymNMF baseline learns independent factors for each timestep and individual, and its reconstruction loss can thus be seen as a lower bound for all the matrix decomposition models we have evaluated. Note that the reconstruction loss increases for all models that explicitly or implicitly add a temporal regularization, which is to be expected. For this data, there is a lot of variation over time that is not necessarily meaningful (we derive interactions from spatial proximity contacts, and therefore there's a lot of randomness in the interaction matrices).
>
> The reconstruction loss of the SymNMF baseline corresponds to a $R^2$ of 0.56, whereas the underlined model in Table 2 has a $R^2$ of 0.35. Please note that we don’t intend to say that all of this variance is meaningless, but we can at least show that our model can better extract the meaningful parts of the variance than the baselines.

---

> > ### Author Response · Authors · 2020-11-25
> > **Response to Reviewer 4 (2/2)**
> >
> > > Lastly, there is a fundamental problem I have with the authors' regularization approach to interpretability. It seems as though they have enforced a handful of individual features for their model in order to achieve interpretability. However, these features are ostensibly things they already know about the individuals in the population. The point of an unsupervised method is to allow yourself the opportunity to discover things that you didn't already know from observing the features of individuals. Moreover, "interpretability" is somewhat subjective. Therefore, I'm concerned that the authors are using these regularization terms a little too haphazardly and have failed to make a strong argument for their use vis-a-vis their effect on reconstruction error.
> >
> > We wholeheartedly agree that interpretability is subjective and can mean different things in different settings. Our main objective in this study was to study a) how much individuality exists in the developmental trajectories of honey bees over their entire lifetimes, and b) how much structure there is in these individual deviations from the "norm." These two aspects are captured by the individuality embeddings and the basis functions, and we have found the model to work without explicit regularization. We have then included the regularization terms to make these learned basis functions (and the resulting lifetime trajectories) easier to understand. For example, we think it is fair to argue that a model that uses two factors is easier to understand than a model that uses eight strongly correlated factors (see the ablation study in the appendix), especially if the two models are similarly good in the quantitative metrics.
> >
> > > I pains me to give this a mediocre score because I do find the work fascinating. The paper requires more principled motivation for the choices the authors made as well as cleaning up the notation.
> >
> > Thank you again for your extensive review. We have reworked the manuscript, improved the notation, and used the additional space available in the revision to motivate our modeling choices better and improve the evaluation.

---

### Official Review · AnonReviewer3 · 2020-10-28
**Could a simpler model be sufficient?**

**Rating:** 5
**Confidence:** 3

**Review:**

The authors propose a factorization method for exploratory analysis in temporal graphs / social networks. If I understand the model correctly, the new idea is to allow individual-specific differences in temporal trajectories (to account, e.g., for difference in age). Some of these ideas might be under the surface in time-shifted tensor decomposition models (linked below), but I do think that the authors' work is novel:

https://www.biorxiv.org/content/10.1101/2020.03.02.974014v2
https://doi.org/10.1016/j.neuroimage.2008.05.062

Though I think the core idea is interesting, I have mixed feelings about the method and some of the analyses.

(1) The method seems potentially more complex than is necessary. Honestly I found its presentation in the paper confusing and difficult to read. Perhaps section 2.1 is a bit too short, and more visualizations like Fig 2 would be helpful.

I think the biggest problem is that the authors' motivation for various choices is not well-described. For example, the adversarial loss is not well-explained, and it is referred to as "L_adversarial" in Fig 2, but "R_adv" in the Appendix (or are these two different things?).

Perhaps a bigger question is why the authors chose to decompose their factors $f_i = m(t_i) + o_i$ (see eq. 3). I think this is an interesting choice, but it seems like there could be many others like $f_i = \sum_j \alpha_{ij} m_j(t_i)$ or $f_i = m(t_i)$. If I understood the paper correctly, the second option is a simpler extension of Yu, Aggarwal, & Wang (2017), which seems like natural to explore before trying the more complicated model proposed in this work.

Smoothness over time seems to be an important aspect of the model (as judged from the schematic figures), but it is not emphasized by the authors how this is achieved in practice. Rather than having a neural network output the mean m, why not use a gaussian process, or a linear layer with a smooth spline basis / RBFs?

(2) If I understand the model correctly, it seems very similar to a classic tensor factorization, so it would seem to me that a better baseline to include would be $\hat{A}_t = F D_t F^T$ where each $D_t$ for $t = 1, ..., T$ is constrained to be diagonal. There are very fast and well-established algorithms for this (see Kolda & Bader, 2009). Also see the time-shifted model variants linked above.

(3) The evaluation metrics are a bit strange. The model performs worse than the baselines on reconstruction, though this does not seem to be the primary goal of the paper. Rather, the idea is to pull out interpretable factors. I had a hard time understanding whether the consistency / mortality / rhythmicity metrics were actually a good proxy for model interpretability. My suggestion would be to focus on qualitative / visual comparisons across the different models rather than these numeric comparisons.

(4) It would improve the paper to show the method working on a second dataset (e.g. a different social network or temporal graph of interactions).

In Summary: I like some of the ideas presented in this work, but I found it difficult to read. I think the biggest problem is that the core idea (individual-specific time courses) seems to get buried and lost with other modeling innovations like adversarial training and new terms like "individuality basis functions." Some of these bells and whistles may not even be needed, so simplifying the model and streamlining the text would go a long way for me.

---

> ### Author Response · Authors · 2020-11-25
> **Response to Reviewer 3 (1/2)**
>
> Thank you for the review, and in particular for the references to the time-shifted tensor decomposition methods. In the revision, we have streamlined the presentation of the main ideas of the paper, cleared up the notation, implemented the constrained tensor decomposition baseline suggested by you, and have used the additional space to justify our modeling choices better:
>
> > (1) The method seems potentially more complex than is necessary. Honestly I found its presentation in the paper confusing and difficult to read. Perhaps section 2.1 is a bit too short, and more visualizations like Fig 2 would be helpful.
>
> This is a valid point of criticism. We have reworked the methods sections and improved the notation, which hopefully makes it easier to follow. In addition to the baselines we evaluate in this work, we have also tested various simpler models, and have found them to not work well on this data.
>
> > I think the biggest problem is that the authors' motivation for various choices is not well-described. For example, the adversarial loss is not well-explained, and it is referred to as "L_adversarial" in Fig 2, but "R_adv" in the Appendix (or are these two different things?).
>
> We have fixed these notational errors and also expanded upon our motivations for the various modeling choices.
>
>
> > Perhaps a bigger question is why the authors chose to decompose their factors (see eq. 3). I think this is an interesting choice, but it seems like there could be many others like or . If I understood the paper correctly, the second option is a simpler extension of Yu, Aggarwal, & Wang (2017), which seems like natural to explore before trying the more complicated model proposed in this work.
>
> We now describe in the methods section why we chose this particular decomposition. Our aim here was to disentangle three things: a) The individuals' average developmental trajectory through their lives (determined by their ages). b) _How_ individuals can deviate from this average trajectory (the basis functions). c) _How strongly_ these possible deviations apply to each individual (the individuality embeddings). If we understood you correctly, the second option is equivalent to the average trajectory which is part of our model, but cannot be used in our evaluation metrics because it is identical for all individuals.
>
> > Smoothness over time seems to be an important aspect of the model (as judged from the schematic figures), but it is not emphasized by the authors how this is achieved in practice. Rather than having a neural network output the mean m, why not use a gaussian process, or a linear layer with a smooth spline basis / RBFs?
>
> Smoothness is not an essential part of the model and arises naturally from the data because transitions in tasks in honey bees tend to occur gradually (the smoothness might be facilitated by the choice of using neural networks to model the trajectories, though). Other choices than neural networks would likely work as well, but this opens up interesting possibilities for future work, e.g., instead of only using the individuals' ages as inputs, we could use multiple inputs (date, age, time of day as unit vectors), in which the networks can learn to generalize the relationship of these inputs even if only sparse data is available.
>
> > (2) If I understand the model correctly, it seems very similar to a classic tensor factorization, so it would seem to me that a better baseline to include would be where each for is constrained to be diagonal. There are very fast and well-established algorithms for this (see Kolda & Bader, 2009). Also see the time-shifted model variants linked above.
>
> Thank you, we have implemented this baseline and added it to the evaluation in the revision. While methodically different, we also believe that the time-shifted tensor decomposition model proposed by Williams 2020 solves a similar problem (even though they analyze spike-trains with multiple trials per neuron, while we have only one observation of each individual in our data, and rather exploit prior knowledge to learn a shared representation). Unfortunately, the tensortools library linked in the preprint does not yet appear to have an implementation of the proposed time-shifted tensor decomposition model, and we were not able to include a fair comparison with this model for this revision.
>
> We also now reference tensor composition models in our introduction.

---

> > ### Author Response · Authors · 2020-11-25
> > **Response to Reviewer 3 (2/2)**
> >
> > > (3) The evaluation metrics are a bit strange. The model performs worse than the baselines on reconstruction, though this does not seem to be the primary goal of the paper. Rather, the idea is to pull out interpretable factors. I had a hard time understanding whether the consistency / mortality / rhythmicity metrics were actually a good proxy for model interpretability. My suggestion would be to focus on qualitative / visual comparisons across the different models rather than these numeric comparisons.
> >
> > The SymNMF baseline learns independent factors for each timestep and individual and it’s reconstruction loss can thus be seen as a lower bound for all the matrix decomposition models we’ve evaluated. Note that the reconstruction loss increases for all models that explicitly or implicitly add a temporal regularization, which is to be expected. For this data, there’s a lot of variation over time that is not necessarily meaningful (we derive interactions from spatial proximity contacts, and therefore there’s a lot of randomness in the interaction matrices). We’ve reworked the manuscript and now better describe why we believe our metrics are meaningful.
> >
> > > (4) It would improve the paper to show the method working on a second dataset (e.g. a different social network or temporal graph of interactions).
> >
> > We have considered this for quite a while but ultimately decided to focus on the honey bee dataset (which is an essential part of the manuscript because we also release the data). We could evaluate similar social network datasets, but we are not aware of any dataset with labels that could improve the evaluation of the method (and interpreting the results for other animals/networks also requires domain knowledge). We are working on applying the method to different datasets (see our response to R1), but we believe that including these ideas would be out of scope for this manuscript.
> >
> > > In Summary: I like some of the ideas presented in this work, but I found it difficult to read. I think the biggest problem is that the core idea (individual-specific time courses) seems to get buried and lost with other modeling innovations like adversarial training and new terms like "individuality basis functions." Some of these bells and whistles may not even be needed, so simplifying the model and streamlining the text would go a long way for me.
> >
> > Thank you again for the suggestions. We have streamlined the text and highlighted that the regularizations, including the adversarial training, are optional and that the model works well even without them (but also show in the ablation study how they improve the model). We also removed the non-negativity regularization because we have found that the model converges well without it.

---

### Official Review · AnonReviewer1 · 2020-10-29
**Interesting paper but limited scope and unclear usefulness --- weak reject**

**Rating:** 6
**Confidence:** 3

**Review:**

# Summary

The authors introduce a novel method for non-negative matrix factorization for timeseries and apply it to longitudinal honey bee interaction data.  The model leverages consistency of individuals over time by forcing the factors (or rather, the residuals of the factors with respect to a global trajectory) to be linear combinations of a small set of temporal basis functions.  These temporal basis functions are functions of the bee’s age.  In other words, the factor embedding of each bee is a vector of linear combinations of 16 learned basis functions of time.  Since all bees use the same 16 temporal basis functions, given these basis functions the lifetime embedding of each bee is encapsulated by a small matrix of numbers, namely the coefficients for the temporal basis functions for each factor (and in practice only two factors were significant, so each bee’s life is embedded in 32-dimensional space).  There are a number of regularizations on the temporal basis functions and the embedding coefficients.

The authors show that the two significant factors have values that vary somewhat consistently as a function of a bee’s age.  The authors also offer some interpretability of the temporal basis functions as corresponding to different social roles in the bee colony.  The authors claim that this interpretable representation of individual bees is valuable for understanding social dynamics in the colony and may be useful for understanding how various perturbations affect the social order of the colony.

# Pros

* The paper is clear and well-written.
* The authors are open-sourcing both their model code and their dataset (which involved a substantial undertaking to collect).
* The authors’ method is novel as far as I know, and it is elegant (albeit has quite a lot of regularization terms).
* The authors are pretty rigorous in their analysis, ablations, and comparison to baselines.

# Cons

I have a few primary concerns, listed here roughly in order of significance:
* I am concerned that this paper might have too narrow a scope in too niche a field for it to be suitable for ICLR.  The paper is entirely dedicated to exploring honey bee social behavior, and it is not clear whether their method would be useful/interesting for other areas of research.  While the authors do suggest that it might be useful for analyzing social behavior in other animals (and make vague references to potential broader applications), they provide no concrete evidence for that.  Without concrete ideas for applications aside from animal social behavior and without strong implications for the usefulness of the representations (see following point), I am concerned that this paper will fail to capture the interest of the vast majority of ICLR attendees.  Perhaps it is better suited for an animal behavior or computational biology journal/conference.
* While the paper focuses on interpretable representations of honey bee behavior, it lacks evidence/support for the usefulness of these representations.  After reading the paper I am left wondering why being able to extract these representations is valuable.  First, in order to extract the representations one must have a longitudinal dataset of tracked honeybee positions.  This I think is only possible to obtain in controlled laboratory settings (though the authors redact the exact method they use), hence the method does not seem broadly applicable.  Second, even if the data were easy to collect, what do the representations indicate that cannot be obtained directly from the data?  If the temporal basis functions merely reflect what hive locations a bee frequents, can that not be extracted much more easily from the tracking data that the model relies on for training?  Third, the authors allude to potential usefulness of the representations for understanding the influence of pesticides or predatory pressure on the colony, but I think the paper really needs a concrete demonstration of (or at least very strong justification for) the usefulness of the representations.
* I definitely appreciate the sweeps over the regularization hyperparameters in table 2, but I am worried about whether these hyperparameters will transfer to other dataset, since the model has quite a number of hyperparameters.  So from a user perspective it would be good to know how robust the authors’ hyperparameters are across datasets.  One way to inform this would be to evaluate the hyperparameter sweeps for the synthetic datasets and see whether optimal hyperparameters (e.g. for the AMI score) are consistent across synthetic datasets.
* Please include the baseline models in Figure 3 (if it is possible to compute AMI for the baselines).  Without baselines, it’s difficult for a reader to determine how good the TNMF curve is.

Finally, I have a few very minor points:
* Please put a legend in the leftmost panel of Figure 4.
* In Figure 5-b, I’m guessing the “f_2” label in the legend supposed to be “f_3”.  If that is true, please change it.  If not, please justify why you are using f_2 in Figure 5 instead of f_3 when it is previously stated that only two factors (f_1 and f_3) are significant.
* The caption for Table 2 reads “Table 3” --- please change that to read “Table 2.”


# Conclusions

I do not recommend accepting this paper.  My main concerns are that the usefulness of the method is not demonstrated or well-justified, and the paper as a whole seems to have limited scope in a field of animal behavior that I suspect is pretty far from most ICLR attendees’ research interests.  However, the AC’s are in a better position than I am to judge the latter point, so I am not certain about my recommendation and could be convinced otherwise.

---

> ### Author Response · Authors · 2020-11-25
> **Response to Reviewer 1 (1/2)**
>
> We truly appreciate the work you've put into this review, and you've given an excellent summary of our proposed method. We've much improved the initial manuscript and hope that we were able to address your concerns:
>
> > I am concerned that this paper might have too narrow a scope in too niche a field for it to be suitable for ICLR. The paper is entirely dedicated to exploring honey bee social behavior, and it is not clear whether their method would be useful/interesting for other areas of research. While the authors do suggest that it might be useful for analyzing social behavior in other animals (and make vague references to potential broader applications), they provide no concrete evidence for that. Without concrete ideas for applications aside from animal social behavior and without strong implications for the usefulness of the representations (see following point), I am concerned that this paper will fail to capture the interest of the vast majority of ICLR attendees. Perhaps it is better suited for an animal behavior or computational biology journal/conference. [...] the paper as a whole seems to have limited scope in a field of animal behavior that I suspect is pretty far from most ICLR attendees’ research interests. However, the AC’s are in a better position than I am to judge the latter point, so I am not certain about my recommendation and could be convinced otherwise.
>
> We initially considered submitting this work to a computational biology journal/conference, but finally decided to submit it to ICLR for the following reasons: Even though we restrict ourselves to the analysis of the honey bee data (and a synthetic baseline) in this work, we do believe that it has applications in other fields, some of which we're currently exploring in follow up work. In particular, the method is trivial to extend to learn factors that are not just a function of time but of multiple (temporal) inputs and even other covariates. Consider the case of a recommender system in which a user's interactions with items might depend on the time (e.g., preferences change over time and new movies are released every year), how long the user has interacted with the platform (e.g., might seek out high profile items initially, and look for 'hidden treasures' later on), and even on the time of day (e.g., might be looking for educational content during the day and entertainment during the night). While such dependencies could also be modeled using higher-order tensor decomposition or deep recommender systems, we think that our formulation is much more suited for such settings. Because the release of the honey bee data is an essential part of this publication, we did decide to focus on this data in this manuscript. In the end, if this is a suitable manuscript for ICLR is something the AC has to judge, but we also want to mention that 'applications in computational biology' are explicitly listed in the ICLR call for papers, and that the main contribution of our work is in representation learning.

---

> > ### Author Response · Authors · 2020-11-25
> > **Response to Reviewer 1 (2/2)**
> >
> > > While the paper focuses on interpretable representations of honey bee behavior, it lacks evidence/support for the usefulness of these representations. After reading the paper I am left wondering why being able to extract these representations is valuable. First, in order to extract the representations one must have a longitudinal dataset of tracked honeybee positions. This I think is only possible to obtain in controlled laboratory settings (though the authors redact the exact method they use), hence the method does not seem broadly applicable. Second, even if the data were easy to collect, what do the representations indicate that cannot be obtained directly from the data? If the temporal basis functions merely reflect what hive locations a bee frequents, can that not be extracted much more easily from the tracking data that the model relies on for training? Third, the authors allude to potential usefulness of the representations for understanding the influence of pesticides or predatory pressure on the colony, but I think the paper really needs a concrete demonstration of (or at least very strong justification for) the usefulness of the representations.
> >
> > Thank you for pointing this is out. We identified several parts of the manuscript that were somewhat misleading and have improved these sections. Most importantly, while a bee's task strongly depends on its age and is reflected in its location, the location distribution is only a rough proxy of its task. In addition to the correlation of the learned factors with location, we also evaluate the representations using the Consistency, Circadianess, and Mortality metrics and are confident that the learned factors represent more than just location. Please note that collecting and analyzing this data is a multi-year process and that experimental treatments we can now perform and analyze using this method can not be included in this manuscript. We have revised the manuscript and hope that we could better highlight why we believe the learned representations are useful. Lastly, while not part of the manuscript, acquiring these datasets requires a controlled setup, but this setup is reproducible, and we already have collaborations with biology groups reusing our setup to produce similar data in their experimental settings.
> >
> > > I definitely appreciate the sweeps over the regularization hyperparameters in table 2, but I am worried about whether these hyperparameters will transfer to other dataset, since the model has quite a number of hyperparameters. So from a user perspective it would be good to know how robust the authors’ hyperparameters are across datasets. One way to inform this would be to evaluate the hyperparameter sweeps for the synthetic datasets and see whether optimal hyperparameters (e.g. for the AMI score) are consistent across synthetic datasets.
> >
> > In the revision, we highlight that the regularizations (and therefore most hyperparameters) are optional and are only meant to increase the interpretability of the results (e.g., to learn a sparse set of a few factors instead of many correlated factors). We have included an unregularized variant of the model in the synthetic dataset evaluation and the main results table.
> >
> >
> > > Please include the baseline models in Figure 3 (if it is possible to compute AMI for the baselines). Without baselines, it’s difficult for a reader to determine how good the TNMF curve is.
> >
> > We have included the SymNMF baseline in Figure 3 and added an additional metric that measures how well the learned factors correspond to the true factors. Unfortunately, our evaluation runs of the other baselines did not finish in time, but the initial results strongly indicate that they perform similarly to the SymNMF baseline, which is consistent with the main results in table 2. We will include these additional baselines in the camera-ready version.
> >
> > > Please put a legend in the leftmost panel of Figure 4.
> >
> > We have now included the legend.
> >
> > > In Figure 5-b, I’m guessing the “f_2” label in the legend supposed to be “f_3”. If that is true, please change it. If not, please justify why you are using f_2 in Figure 5 instead of f_3 when it is previously stated that only two factors (f_1 and f_3) are significant.
> >
> > We’ve renamed the two factors that the model uses to f_0 and f_1 and adjusted all references to these factors accordingly.
> >
> > > The caption for Table 2 reads “Table 3” --- please change that to read “Table 2.”
> >
> > Thank you, we have fixed the label.

---

### Official Review · AnonReviewer2 · 2020-10-31
**The learned individual embeddings need additional clarification**

**Rating:** 5
**Confidence:** 4

**Review:**


This paper proposes a NMF formulation ||A-FF^T||^2 where A and F are different types of information extracted from social datasets. In the honeybee example the authors highlight, A represents the spatial relationship between bees, and F encodes the age of the bees. The authors setup F to be decomposable into two types of embeddings, one which characterizes the group activity and the other which characterizes the individual activity.

Pros:
The authors highlight a very interesting application. Honey bees perform tasks according to their age (information in F) and the tasks are performed in different nest substrates (information in A).


Cons:
 It is not clear what is the form of the embeddings e_\phi.
It is not clear what is the baseline for the interpretable individual embeddings. Why should these individual embeddings capture reasonable offsets and why they capture the information in c_{i,t}? This is only tested indirectly, through the AMI score in the case of the synthetic dataset. Is there a way to setup this framework to check that true F and learned F in the synthetic dataset match more directly? In the current version of the work I cannot tell how well F captures the information encoded in c_{t,i}, this makes it hard to consider why/if ||A-FF^T||  could be meaningful.
What is the factor strength in Fig 4?
The variance of AMI score is high even when the SNR = inf. Do the authors consider the model results robust given this high variance in the synthetic dataset?

Comments:
Is there a mismatch in the type of data of Y in the synthetic and in the true datasets?  In the synthetic dataset is Y the spatial distance like in the honeybee dataset?
The notation in Eq 3 is not clear, c: \mathbb{N}^{TxN} -> \mathbb{N}, but o_{\phi, \omega} \mathbb{N}^{Nx T} -> \mathbb{R}^M? Why are the inputs in \o_{} NxT?
The authors use the terms functional position, functional role, and functional embedding. The way the paper is written it seems like these terms could be used interchangeably, but isn’t this is what the authors are trying to show in the paper? It would be easier to understand this work if the authors explained in more detail the different variables/mappings.

---

> ### Author Response · Authors · 2020-11-25
> **Response to Reviewer 2**
>
> We want to thank you for the review and the points of criticism that we have taken into account for the revision of the manuscript.
>
> Please note that F is supposed to not only take the age of an individual into account, but also how it deviates from the common developmental trajectory. We capture these deviations via the learned basis functions $b_k(\textrm{age})$, and the embeddings $\phi$ describe to what extent each basis function applies to each individual. We've much improved the notation and also now more extensively justify our modeling choices.
>
> Following is a point-by-point response to your comments:
>
> > It is not clear what is the baseline for the interpretable individual embeddings. Why should these individual embeddings capture reasonable offsets and why they capture the information in c_{i,t}? This is only tested indirectly, through the AMI score in the case of the synthetic dataset. Is there a way to setup this framework to check that true F and learned F in the synthetic dataset match more directly? In the current version of the work I cannot tell how well F captures the information encoded in c_{t,i}, this makes it hard to consider why/if ||A-FF^T|| could be meaningful.
>
> Thank you for the suggestion. We now also evaluate how well the learned factors F match the ground truth factors in the synthetic dataset and compare them to the SymNMF baseline. For the honey bee dataset, while this is an unsupervised learning problem, we have some labels that we use in the Consistency, Mortality, and Rhythmicity metrics. These metrics are supposed to verify that the individual embeddings are meaningful. Please note that these metrics depend on labels extracted from the raw honey bee movement data, not from the interaction networks themselves, and that those labels do capture biologically relevant properties of the individuals.
>
> > What is the factor strength in Fig 4?
>
> We've improved the figure in the revision. The factor strength on the left side corresponds to the common trajectory of factors ($m(\textrm{age})$), and the factor strength on the right side corresponds to the individual factors F.
>
> > The variance of AMI score is high even when the SNR = inf. Do the authors consider the model results robust given this high variance in the synthetic dataset?
>
> We've found that the high variance was caused by multiple reasons: a) Convergence issues. We've increased the number of iterations and batch size and have found that this improved the results considerably. b) Correlated factors: Usually, there are multiple possible factorizations, especially when the truth factors are correlated. c) We've also added a model without the interpretability regularizations, and found that in the case of low noise, it performed better than the regularized variant. Note that in the high-noise settings (which are more similar to our dataset), the two variants have a similar performance. We also added the SymNMF baseline to put these numbers into context and found that our model performs much better.
>
> > Comments: Is there a mismatch in the type of data of Y in the synthetic and in the true datasets? In the synthetic dataset is Y the spatial distance like in the honeybee dataset?
>
> For the synthetic data, we don't create movement trajectories but rather create a trajectory of ground truth factors that we use to generate simulated interaction matrices. We've improved the description in the paper to make this clearer.
>
> > The notation in Eq 3 is not clear, c: \mathbb{N}^{TxN} -> \mathbb{N}, but o_{\phi, \omega} \mathbb{N}^{Nx T} -> \mathbb{R}^M? Why are the inputs in \o_{} NxT?
>
> Thank you. We've improved the notation throughout the manuscript.
>
> > The authors use the terms functional position, functional role, and functional embedding. The way the paper is written it seems like these terms could be used interchangeably, but isn’t this is what the authors are trying to show in the paper? It would be easier to understand this work if the authors explained in more detail the different variables/mappings.
>
> Yes, we want to show that the learned factors and embeddings are useful representations of the individuals' functional roles in the colony. We agree that the notation and terms used were somewhat confusing and inconsistent and have improved them throughout the manuscript.

---

### Decision · Program_Chairs · 2021-01-07
**Final Decision**

**Decision:**

Reject

**Comment:**

The authors present a matrix factorization for the social behavior of honey bees in a hive. All the reviewers appreciated the interesting application.  However, substantial concerns were raised about the model motivation and the interpretation of the learned factors. To quote one reviewer, "Some of these bells and whistles may not even be needed, so simplifying the model and streamlining the text would go a long way for me." Another said, "The paper requires more principled motivation for the choices the authors made as well as cleaning up the notation."  The authors did address some of these concerns in discussion, but there are too many lingering concerns to recommend acceptance.  Given the unique application of this paper, the authors might also consider a journal that specializes in computational biology instead.